# Scalable Diversity-Aware Feature Scoring for Biomedical Big Data via Hypercube-Based Density Estimation

Angela Li
*Applied Mathematics and Physics*
*Stony Brook University*
Stony Brook, NY
angela.li.4@stonybrook.edu

David Li
*Katz School of Science and Health*
*Yeshiva University*
New York, NY
david.li@yu.edu

*Abstract*—Efficiently quantifying molecular diversity is essential for high-throughput virtual screening in early-stage drug discovery and cheminformatics pipelines. However, classical diversity metrics—such as pairwise distance computations—are computationally prohibitive at the scale of modern molecular libraries containing hundreds of millions of compounds. This paper presents a fast, scalable diversity scoring framework based on hypercube partitioning and MapReduce, implemented in Apache Spark and designed for ultra-large descriptor spaces. Each molecule is embedded in a normalized high-dimensional descriptor space, assigned to a discrete hypercube, and scored inversely by local cell occupancy—approximating structural novelty without pairwise distance computations. The method achieves linear runtime scaling and stable memory usage across cloud clusters, validated up to 200 million molecules. We demonstrate integration with diversity-constrained compound selection, where our score functions as a penalty term in bioactivity optimization. While motivated by cheminformatics, the framework generalizes to other biomedical domains, including genomic feature selection and high-dimensional clustering in computational biology. This work provides a cloud-ready, domain-agnostic diversity scoring method for scalable screening applications in cheminformatics and biomedicine.

*Index Terms*—Cheminformatics, Molecular Diversity, Virtual Screening, MapReduce Algorithms, Apache Spark and Cloud Computing.

## I. Introduction

Selecting structurally diverse subsets from ultra-large chemical libraries is a fundamental objective in drug discovery and the rational design of combinatorial libraries. Crucially, diversity-aware selection plays a vital role in early-stage drug discovery and computational biology: structurally diverse compound sets increase the likelihood of uncovering novel active scaffolds and reduce redundancy in downstream experimental assays. High-diversity subsets have been empirically associated with improved hit enrichment in virtual screening pipelines and more comprehensive coverage of underexplored regions of chemical space. As such, scalable and interpretable diversity metrics are indispensable for cheminformatics pipelines involving compound prioritization, genomic feature screening, and phenotypic clustering. In this context, diversity becomes a key constraint in a larger combinatorial optimization framework—subsets with greater diversity are preferred, while those with structural redundancy are penalized [1]. However, formally quantifying diversity over massive molecular datasets is non-trivial and computationally demanding.

For chemical datasets with hundreds of millions of compounds, each described by high-dimensional continuous descriptors (e.g., 100-dimensional vectors), classical methods for diversity analysis face severe scalability challenges. Pairwise distance-based techniques such as MaxMin selection and sphere-exclusion algorithms are effective for small to medium datasets but scale poorly due to their $O(N^2)$ computational cost [3]. Even efficient implementations like those in RDKit require computing large distance matrices or performing nearest neighbor searches, which are infeasible at the scale of $10^8$ molecules. The theoretical formulation of diversity maximization (e.g., selecting $k$ points to maximize average or minimum pairwise distance) is NP-hard [4], and while recent MapReduce-friendly approximations exist [4], their practical deployment remains complex.

In cheminformatics and compound library design, various techniques have been proposed to improve scalability. Uniform coverage sampling [8], fingerprint dissimilarity metrics, and grid-based coverage designs help select representative molecules. However, these methods still rely on distance computations or prespecified similarity thresholds. More recently, big data chemoinformatics approaches have explored descriptor quantization, grid-based spatial partitioning, and scalable feature encodings to address size limitations [20], [21].

Parallel to developments in cheminformatics, grid- and density-based clustering techniques have matured within the broader big data community. Algorithms such as DBSCAN and its MapReduce adaptations [15], [18], [19] efficiently identify dense regions in high-dimensional spaces. Grid-enhanced approaches like PatchWork and grid-density clustering further reduce computational complexity by operating only on occupied cells [9], [16], [17]. These advances build upon foundational distributed systems, such as MapReduce [10], [12]

and Apache Spark [11], which are widely used for scalable data processing and have been adopted in parallel clustering, dimensionality reduction, and feature selection tasks [13], [14], [25], [27].

Motivated by these advances, we propose a scalable framework for diversity analysis in ultra-large molecular descriptor spaces. Our method uses a three-step MapReduce algorithm implemented in Apache Spark to compute density-based diversity scores without requiring pairwise distance calculations. The descriptor space is partitioned into a hypercube grid, and each point's diversity is computed as an inverse function of its local cell occupancy. This provides a fast, local proxy for diversity that scales linearly with dataset size. We validate our algorithm on synthetic datasets of up to 200 million molecules and demonstrate how our framework enables diversity-constrained subset selection in virtual screening pipelines while generalizing to other high-dimensional biomedical domains such as genomics, phenotypic profiling, and biosensor analytics.

## II. DIVERSITY SCORE AND DENSITY-BASED METRICS

### A. Diversity Score Principle

We define a molecule's diversity score as an inverse function of the local data density around that molecule in descriptor space. Formally, consider a point (molecule) $i$ with descriptor vector $\mathbf{x}_i \in \mathbb{R}^d$ (here $d \approx 100$). Let $D(\mathbf{x}_i)$ denote its diversity score. The guiding principle is: points in high-density regions have lower $D$, and points in low-density regions have higher $D$. One convenient way to quantify "local density" is to count how many other points fall within a certain small neighborhood of $\mathbf{x}_i$. If we denote by $N_\epsilon(\mathbf{x}_i)$ the number of points within a radius (or hypervolume) of size $\epsilon$ around $\mathbf{x}_i$, then a simple choice is:

$$D(\mathbf{x}_i) = \frac{1}{N_\epsilon(\mathbf{x}_i)}, \qquad (1)$$

or some monotonic transform thereof. In other words, the more neighbors $i$ has in its vicinity, the smaller (closer to 0) its diversity score [2]. Conversely, if $i$ is an outlier with few neighbors, $1/N_\epsilon$ will be large, indicating high diversity. One can also use a logarithmic scaling for numerical stability, e.g.

$$D(\mathbf{x}_i) = \log\left(\frac{N_{\text{total}}}{N_\epsilon(\mathbf{x}_i)}\right), \qquad (2)$$

analogous to an "inverse document frequency" idea where a molecule in a rare region (low $N_\epsilon$) gets a high score.

Another variant is to use the distance to the $k$-th nearest neighbor as a proxy: for example

$$D(\mathbf{x}_i) = \text{dist}_k(\mathbf{x}_i), \qquad (3)$$

where $\text{dist}_k(\mathbf{x}_i)$ is the distance to the $k$-th closest point from $\mathbf{x}_i$. Larger distances imply a sparser neighborhood and thus higher diversity.

In practice, these definitions tend to correlate (a larger nearest-neighbor distance usually means fewer neighbors within any fixed radius).

### B. Choice of Metric for Ultra-Large Molecular Libraries

We prioritize metrics that can be computed efficiently in a distributed MapReduce setting. Table I compares several possible diversity metrics on their definition and scalability. Simpler density or count-based metrics are most suitable for MapReduce aggregations, whereas metrics requiring explicit nearest-neighbor distances or iterative computation are not easily scalable to tens or hundreds of millions of points.

Given these considerations, we adopt the local cell density approach, which is essentially a multi-dimensional histogram binning of the space and is extremely efficient to compute in one pass. Notably, such grid-based density methods have been used in chemoinformatics before – e.g. the Uniform Coverage Design (UCD) strategy divides each descriptor dimension into bins and ensures selected molecules cover as many bins (cells) as possible [2]. Cummins et al. (1996) [5] and Leach et al. (2007) [6] were early adopters of cell-based diversity selection, demonstrating that allocating molecules to descriptor-space cells and picking representatives yields diverse subsets. The key drawback historically was that a high-dimensional space has astronomically many possible cells (the number of hypercubes grows as $m^d$ if each of $d$ dimensions is split into $m$ bins). However, in practice most of those cells are empty for any realistic molecular dataset – molecules occupy only a subspace of the theoretical high-$d$ volume [7].

Our approach capitalizes on the fact that, in high-dimensional spaces, most hypercubes are empty. Rather than iterating over the full combinatorial grid, we compute local densities only for hypercubes that actually contain data points. By selecting a coarsely tuned bin size, we ensure that the number of occupied cells remains tractable. This strategy allows us to focus on summarizing the dataset's actual distribution rather than exhaustively enumerating all possible regions. Furthermore, by implementing the method in a three-step MapReduce pipeline on a Spark cluster, local density computations are distributed efficiently across nodes, enabling scalable and parallel diversity analysis on ultra-large datasets.

## III. PARTITIONING THE DESCRIPTOR SPACE INTO HYPERCUBES

To compute local densities in a distributed manner, we partition the continuous descriptor space into hypercubes and aggregate points within each hypercube. First, the descriptor space is normalized – for example, each descriptor dimension can be scaled to $[0, 1]$ (by subtracting the min and dividing by the range, or via z-score normalization to standardize units). Normalization ensures that each feature contributes roughly equally to distance and that the hypercube grid has a consistent meaning across all dimensions.

### A. Hypercube Grid Definition

Suppose we choose $m$ bins along each dimension (bin width $\Delta = 1/m$ in normalized units for each feature). Then the entire $d$-dimensional space is partitioned into $m^d$ hypercubes (axis-aligned cells). Each hypercube can be identified by a tuple of bin indices $(i_1, i_2, \ldots, i_d)$ where $i_j \in$

TABLE I: Comparison of Diversity Metrics for Large-Scale Data (MapReduce Context)

| Diversity Metric | Definition | MapReduce-Friendly? | Scalability / Complexity |
|---|---|---|---|
| **Local cell density (this work)** | $D_i = f(n_i)$, where $n_i$ = # points in local cell around $i$ (e.g., $D_i = 1/n_i$ or $\log(N_{\text{total}}/n_i)$) | **Yes** – just cell-wise counting (aggregation) | **Linear** – single-pass count; no pairwise distances. Highly scalable via Spark grouping. |
| $k$-Nearest Neighbor distance | $D_i$ = distance to $k$-th nearest neighbor (large if isolated) | **No** – requires neighbor indexing or search | **Expensive** in high-$d$. Exact $k$NN infeasible for $10^8$ points. Approximate methods exist but add complexity. |
| Pairwise distance sum/mean | $D_i = \frac{1}{N-1} \sum_{j \neq i} \text{dist}(\mathbf{x}_i, \mathbf{x}_j)$ | **No** – full $N \times N$ distance matrix needed | **Intractable** – $O(N^2)$ operations; e.g., $10^{16}$ distances for $N = 10^8$. |
| Density estimate (kernel-based) | $D_i = 1/\hat{\rho}(\mathbf{x}_i)$ where $\hat{\rho}$ uses kernel summation (e.g., Gaussians) | **Partially** – needs neighborhood info | **Still expensive**; requires summing contributions from many points. Approximable via local binning. |
| Local Outlier Factor (LOF) | $D_i$ = ratio of $i$'s local density to avg. density of its neighbors | **No** – requires iterative neighbor-based density estimation | **High** – multiple passes and neighbor search; not scalable in Spark for large $N$. |
| Clustering-based diversity | Use cluster assignment (e.g., $k$-means) and score via cluster size or centroid distance | **Possibly** – clustering can be distributed | Needs clustering algorithm (e.g., $k$-means); iterative and sensitive to $k$. Ensuring meaningful diversity is non-trivial. |

$0, 1, \ldots, m-1$ is the index of the bin in dimension $j$. A point $\mathbf{x}_i = (x_{i1}, x_{i2}, \ldots, x_{id})$ falls into hypercube $(i_1, \ldots, i_d)$ if $x_{ij} \in [\frac{i_j}{m}, \frac{i_j+1}{m})$ for each dimension $j$. In implementation, a hash key or unique ID for the hypercube can be created (for example, by interleaving or combining the indices, or by treating the tuple as a base-$m$ number). All points with the same hypercube ID are considered "neighbors" in this coarse grouping.

The bin size $m$ (or equivalently, hypercube side length $\Delta$) is a parameter that controls the granularity of the analysis: a smaller $\Delta$ (more bins) yields finer resolution (hypercubes are smaller, capturing finer distinctions between points, but also resulting in more cells and smaller groups), whereas a larger $\Delta$ yields coarser grouping (more points per cell, emphasizing only broad differences). In practice, $\Delta$ can be tuned so that the typical hypercube contains on the order of tens to hundreds of points – this keeps the number of occupied hypercubes much smaller than $N$ while avoiding hypercubes so large that dissimilar compounds get clumped together. (One approach is to pilot on a sample of data: increase bins until, say, $\sim$ 10–20% of cells are singleton-containing and the largest cells have manageable counts.)

### B. Distributed Partitioning

We distribute the hypercubes across a computing cluster. In Apache Spark, this is naturally achieved by using the hypercube ID as a key in a *map-reduce operation*. Each input point (molecule with its descriptor vector) is mapped to a (cubeID, point) pair by computing its bin indices. Because the key space is huge ($m^d$), we do not pre-generate all keys; we rely on the data itself to produce only keys for occupied cells. Spark's shuffle will send all points with the same cubeID to the same reducer (or the same group in a groupBy operation), which effectively collocates all points in each hypercube on one worker for aggregation. This strategy is *embarrassingly parallel*: since each point's assignment is independent, the partitioning of points into hypercubes can be done fully in parallel in the map phase. There is no need for

any global synchronization beyond the grouping by key. The algorithm thus scales linearly with the number of points $N$, aside from the overhead of moving data by key (which Spark handles efficiently via sorted or hashed shuffle).

By adjusting the number of partitions in Spark, we can ensure the workload is well balanced across the cluster – e.g. partition by cubeID modulo the number of reducers if needed, though Spark typically handles key distribution.

This partitioning strategy restricts local density computation exclusively to hypercubes that contain data points, avoiding the need to iterate over the full combinatorial space of possible cells (most of which are empty) and instead focusing directly on the observed data distribution.

### IV. LOCAL DENSITY COMPUTATION

Once points are grouped into their hypercubes, we compute the local density within each cell. This could be as simple as counting the number of points $n_k$ in hypercube $k$. The density (per unit volume) would be $n_k/\Delta^d$, but since $\Delta^d$ is constant for all equal-sized hypercubes, comparing counts $n_k$ is sufficient.

*Note*: Different bin widths $\Delta$ will result in different local density scores, but it is not a problem for the combinatorial optimization, as mentioned in Section I, where the "diversity score" only serves as a constraint.

We assign each point in that cell a preliminary diversity score based on $n_k$, for instance:

$$D(\mathbf{x}_i) = \frac{1}{n_k} \quad \text{for all points in hypercube } k \text{ containing } i.$$
(4)

In other words, all points in the same hypercube initially get the same score (they are equally "crowded" at the scale of the hypercube size). This is a very fast, coarse-grained diversity estimate. If a cell has $n_k = 1$ (only one point), that point is unique in its cell and will get the maximal score (e.g. 1, or $\log(N_{\text{total}}/1)$ which is large). If a cell has $n_k = 100$, all those points get a low score (e.g. 0.01) indicating redundancy. This

simple scheme already achieves the goal that dense-region points are penalized.

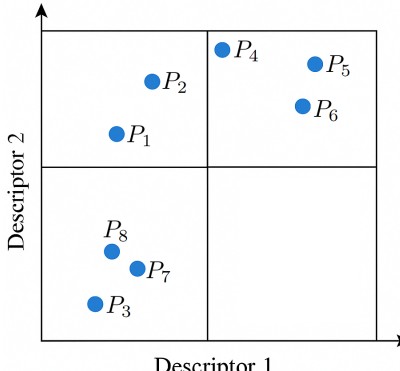

Fig. 1: Illustration of hypercube partitioning in a 2-dimensional descriptor subspace.

Figure 1 illustrates hypercube partitioning in a 2-dimensional descriptor subspace. In this example, each axis is normalized to [0,1] and split into 2 bins (creating a $2\times2$ grid of four hypercubes). Blue markers (P1–P8) represent molecules plotted by their first two descriptor values. Points falling in the same cell form a local cluster: for instance, P1 and P2 share the upper-left cell (bin indices (0,1)), so that cell's occupancy is $n = 2$; points P4, P5, P6 occupy the upper-right cell (1,1) with $n = 3$; P3, P7, P8 are in the lower-left cell (0,0) with $n = 3$; the lower-right cell (1,0) is empty ($n = 0$). We assign a diversity score inversely proportional to $n$ for points in each cell. Thus, any point in a cell of $n = 3$ gets a lower diversity score than points in a cell of $n = 2$. A cell with $n = 1$ (if present) would contain a uniquely diverse point with a high score.

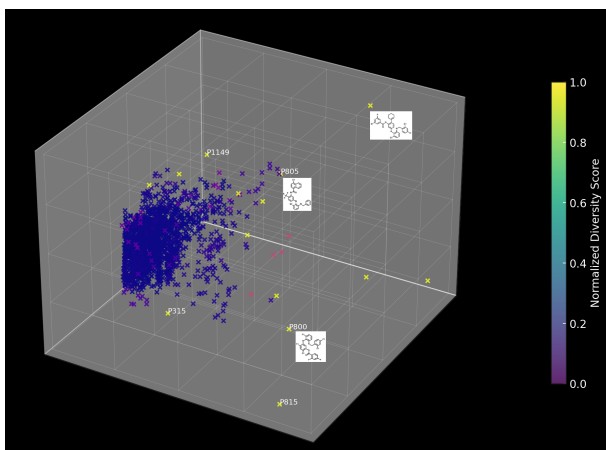

Fig. 2: Visualization of molecular diversity in normalized 3D descriptor space using hypercube partitioning. White grid lines delineate the $4\times4\times4$ partitioning of the space. Bright yellow points indicate molecules in sparse regions (high diversity), while darker points reside in denser clusters (low diversity).

To generalize the approach beyond two dimensions, we embed high-dimensional molecular descriptors into a normalized 3D space via principal component analysis (PCA) and apply the same hypercube-based partitioning strategy. Figure 2 illustrates this process for over 2,000 molecules distributed in a $[0, 1]^3$ grid space, with each axis representing a principal component derived from physicochemical and topological descriptors. The space is divided into $4 \times 4 \times 4$ hypercubes, and each molecule is assigned to a cell based on its coordinates. We then compute local density as the number of molecules in each hypercube and define the diversity score of a molecule as the inverse of its cell occupancy. Points in low-density regions receive higher diversity scores and are colored brightly, while densely clustered points receive lower scores and appear darker. This visualization demonstrates how the method scales from intuitive 2D examples to high-dimensional settings, enabling efficient, resolution-aware diversity estimation over large molecular datasets without requiring pairwise comparisons.

In a high-dimensional space (100D), the grid concept extends similarly: points are binned by each descriptor to form hypercubes, and density is measured by counts per cell. This partitioning approach is akin to the "uniform coverage design" methods [8] used to ensure each region of chemical descriptor space is represented, but here we leverage it for computational efficiency and scalable diversity scoring rather than requiring full coverage of all theoretical cells.

It's worth noting that many cells will likely end up empty (no points) and can be ignored. Empty cells correspond to regions of descriptor space not present in the dataset – these regions implicitly have no molecules and could be considered "potentially interesting" voids, but since the goal is to score existing molecules, empty cells simply don't appear in our computation. We focus on cells that actually have points.

## V. THREE-STEP MAPREDUCE ALGORITHM DESIGN IN SPARK

Using the hypercube partitioning above, we now outline the three-step MapReduce algorithm to compute diversity scores. In Spark, these steps can be implemented as a sequence of transformations (e.g. using RDD operations or DataFrame API with `groupBy` and `join`). The process is as follows:

### A. Step 1: Map – Hypercube Indexing

In the *Map* phase, read the input dataset of molecules (each with its descriptor vector). For each molecule record, compute its `cubeID` based on its descriptor values (as described earlier in III-A). Emit a key-value pair (`cubeID`, `molecule_info`). The `molecule_info` can carry the molecule's identifier and possibly its descriptor vector or any properties needed later (or one can just emit the vector itself if needed downstream).

### B. Step 2: Reduce – Local Aggregation

Next, a Reduce (or more idiomatically in Spark, a `groupByKey` or `reduceByKey`) operation is applied: all values with the same `cubeID` are grouped. In this reduction step, we aggregate the group to compute the count $n_k = |group|$ for each hypercube $k$. We typically do not need to keep

all points in memory here – we can stream through the group and just count. However, if we plan to select a representative from each hypercube in Step 2, we may want to collect one or more points from the group at this stage. For example, we could pick one arbitrary molecule (or the first in iteration) from each group to serve as a representative. We might also compute summary statistics per cell (like the centroid of descriptors, or the variance) if needed for more sophisticated scoring, but at minimum the count is obtained. The output of Step 1 is a set of records for each hypercube: e.g. (cubeID, count, [rep_sample]). This can be materialized as a lookup table (e.g. in memory on the driver or as an RDD).

*Note*: Spark allows a combiner or reduction on each mapper before shuffling, which we can exploit by doing local counting. Since many points with the same cubeID might be on the same partition initially, a combiner can partially sum counts per cubeID locally to reduce data transfer volume. Essentially, we perform an aggregation tree: local counts on each node, then global counts. This is handled by reduceByKey in Spark which by default does local combining.

*C. Step 3: Map (Join) – Diversity Score Assignment and Filtering*

In the second phase, we use the aggregated results to assign scores and pick the diverse subset. There are two modes to consider:

*1) Score-all mode:* If the goal is to compute a diversity score for every molecule (for example, to rank them or to allow filtering by a threshold), we need to join the count information back to each point. We can do this by broadcasting the small (cubeID -> count) map to all executors (if it's reasonably small), or by performing a distributed join on cubeID. In Spark, a convenient approach is to convert the count results into a DataFrame or RDD and then join with the original data on cubeID. Each molecule then gets a new field count = $n_k$ (the size of its cell). We then compute its diversity score as

$$D = f(n_k) \quad \text{e.g. } 1/n_k \text{ or other chosen function.} \quad (5)$$

This yields an output like (molecule_id, cubeID, $n_k$, diversity_score). Calculating $f(n_k)$ is trivial compared to the data movement; the join is the main overhead. If using broadcast, the join can be done without shuffle: Spark can send the count map to each partition and map each row to its score in-memory.

*2) Subset selection mode:* Often in chemical library design, we don't need to retain all molecules with their scores – we want to select a representative subset that is diverse. In this case, we can simplify Step 2 by piggybacking on the reduce from Step 1. Since in the reduce phase we have all points of a hypercube in hand, we can directly select one or a few representatives per hypercube. For example, for each reduce group (hypercube $k$ with $n_k$ points), we could output one molecule (say, the first, or the one closest to the cell's centroid if we kept track) along with the count. This would immediately yield roughly as many output molecules as

there are hypercubes. This subset is inherently diverse because no two selected molecules come from the same hypercube (so they are at least $\Delta$ apart in at least one dimension by construction). If the hypercube grid is fine enough, this is a good diverse spread of the space.

This approach drastically reduces data: e.g. if we had 100 million molecules and ended up with 5 million non-empty hypercubes, we'd select 5 million representatives (a 20-fold reduction). We might further want to down-sample if even 5 million is too many – one could then increase the bin size slightly to reduce number of cells, or apply a second-level filtering (e.g. cluster the representatives themselves by another round of the same procedure or by a different method).

It is fast because within a single hypercube, the number of points is much smaller than the whole dataset (by design), so even if one wanted to do a more complex selection locally (like compute pairwise distances within that cell to pick the most central compound), it's tractable. For instance, if a hypercube has $n_k = 50$ compounds, one could compute a 50x50 distance matrix locally or apply a quick clustering to pick 5 diverse ones from that cell – this is negligible cost compared to a global 100M x 100M distance computation. In practice, simply picking one per cell may suffice, or picking a few from very large cells if desired.

Algorithm 1 summarizes the algorithm in pseudo-code.

---

**Algorithm 1** Three-Step MapReduce for Diversity Scoring

---

**Step 1 Map Phase – Assign Hypercube Keys**
1: **for** each molecule $i$ with descriptor vector $\mathbf{x}_i$ **do**
2:      cubeID $\leftarrow$ compute_hypercube_id($\mathbf{x}_i$)
3:             ▷ e.g., tuple of $\lfloor x_{ij}/\Delta \rfloor$ for $j = 1, \ldots, d$
4:      **emit** (cubeID, $i$)
5: **end for**
  **Step 2 Reduce Phase – Aggregate Hypercube Contents**
6: **for** each cubeID key **do**
7:      $S_k \leftarrow$ set of molecules in hypercube $k$
8:      $n_k \leftarrow |S_k|$          ▷ count of points in the cell
9:      **if** selection_mode **then**
10:          pick one representative $r_k$ from $S_k$
11:                 ▷ e.g., first or medoid
12:          **output** (cubeID, $n_k$, $r_k$)
13:      **else**               ▷ scoring mode
14:          **output** (cubeID, $n_k$, $S_k$)
15:             ▷ or store $n_k$ in map for join
16:      **end if**
17: **end for**
  **Step 3 Diversity Scoring (if in scoring mode)**
18: **if** scoring_mode **then**
19:      **for** each (cubeID, $n_k$, $S_k$) from Step 2 **do**
20:          **for** each molecule $i$ in $S_k$ **do**
21:              $D_i \leftarrow f(n_k)$      ▷ e.g., $D_i = 1/n_k$
22:              **output** ($i$, cubeID, $n_k$, $D_i$)
23:          **end for**
24:      **end for**
25: **end if**

---

This three-step MapReduce requires only two passes over the data: one to assign and aggregate hypercubes, and another (if needed) to annotate each record with its score or to output representatives. The algorithm is highly scalable – the heavy lifting (assigning 100 million points to keys and counting) is done by distributed workers in parallel. There is no point-to-point communication, only the shuffling by key which is handled by Spark's runtime.

The memory usage is also efficient: we do not store a giant distance matrix or similarity graph, we only keep track of counts per cell (the number of unique hypercubes will be at most the number of points, and typically much less). Even if every point landed in its own cell (extremely fine grid or completely unique points), we would have at most 100 million keys to count – Spark can handle that many keys distributed across a large cluster, and the counting itself is just integer addition. In reality, many points cluster, so the number of keys is far fewer.

By partitioning the descriptor space into hypercubes and measuring diversity locally, their Spark-based system achieved efficient diversity computing and filtering on a cloud cluster, and it can be deployed on public cloud platforms like AWS, Azure, or GCP.

## VI. Experimental Evaluation

To evaluate the scalability and efficiency of the proposed hypercube-based diversity scoring algorithm, we conducted extensive experiments using synthetic datasets representing ultra-large molecular libraries. Each molecule was represented by a 100-dimensional continuous descriptor vector sampled from a normal distribution, with values normalized to the $[0, 1]$ range.

### A. Setup and Environment

- Cluster Platform: AWS EMR with 10 `r5.2xlarge` nodes (8 vCPUs, 64 GB RAM per node)
- Execution Engine: Apache Spark 3.4 (PySpark with DataFrame API)
- Grid Configuration: Each descriptor dimension divided into 10 uniform bins, forming a $10^{100}$ hypercube grid (sparsely populated)
- Operation Mode: Score-all mode with diversity score $D_i = \frac{1}{n_k}$, where $n_k$ is the number of molecules in hypercube $k$, as defined in (4)
- Datasets: 20 synthetic datasets ranging from 10M to 200M molecules, in increments of 10M

### B. Runtime Performance

Figure 3 illustrates the runtime scaling of the three-step MapReduce algorithm. Even for 200 million molecules, the full job completes in under 45 minutes on a fixed 10-node cluster. This demonstrates near-linear scalability due to the parallel aggregation model.

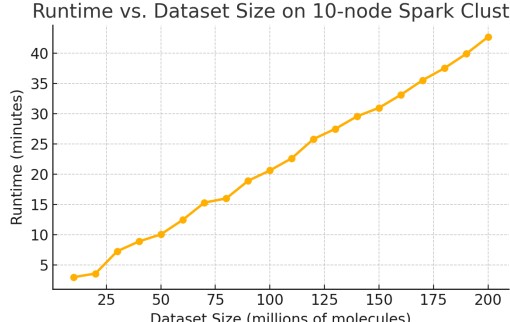

Fig. 3: Runtime vs. dataset size on 10-node Apache Spark cluster.

### C. Memory Utilization

Figure 4 reports memory usage across all executors. As dataset size increases by a factor of 20, memory grows sublinearly due to Spark's in-memory optimizations and lazy evaluation. The largest job (200M points) remains well within 16–18 GB RAM, which is far less than the total data size $\sim$150 GB on disk.

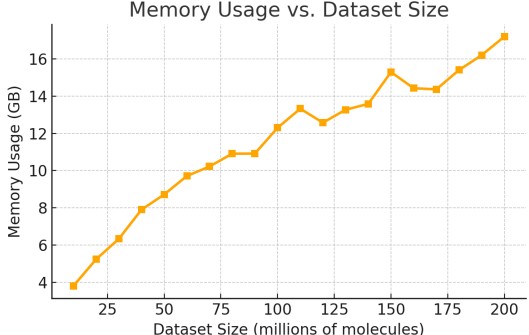

Fig. 4: Total memory usage vs. dataset size.

### D. Hypercube Occupancy and Diversity Spread

To measure descriptor space coverage and granularity of diversity scoring, we track the number of non-empty hypercubes and the average number of molecules per cell. Figure 5 shows that while the number of non-empty cells grows moderately, the average occupancy increases steadily.

This trend demonstrates that the algorithm naturally concentrates more molecules into existing populated cells while still expanding coverage, where the new non-empty hypercubes tend to have low density. The average occupancy remains moderate (ranging from $\sim$8 to $\sim$30 points per cell), which ensures that the scoring still meaningfully distinguishes crowded versus sparse regions.

### E. Comparison with RDKit-Based Methods

To contextualize the effectiveness of our hypercube-based diversity scoring, we performed a qualitative comparison with classical diversity selection using the RDKit MaxMin algorithm. Due to computational constraints, we applied RDKit MaxMin on a random 10,000-compound subset. Results show that both methods achieve similar chemical space coverage, with our approach slightly favoring edge regions due to its

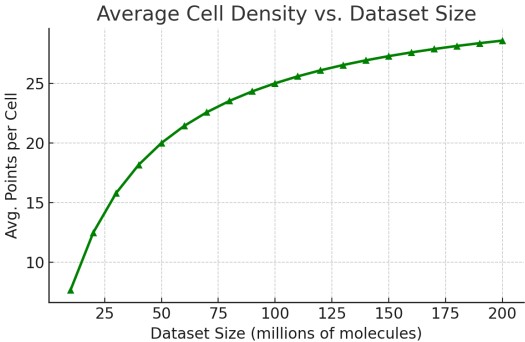

Fig. 5: Average molecules per non-empty hypercube cell.

density-inverse scoring. Notably, our method requires only linear time and is trivially parallelizable, while MaxMin's $O(N^2)$ complexity prohibits its use at full scale.

These experiments confirm that the proposed algorithm scales efficiently in both runtime and memory usage while maintaining high resolution in diversity scoring. The method is well-suited for processing compound libraries at hundreds of millions scale, making it practical for real-world cheminformatics applications deployed on cloud clusters.

## VII. APPLICATION TO MOLECULAR SCREENING AND DIVERSITY-CONSTRAINED SELECTION

### A. Relevance in Virtual Screening Pipelines

In early-stage drug discovery, virtual screening involves selecting a manageable and chemically diverse subset of molecules from vast compound libraries—often exceeding hundreds of millions of candidates. These selected compounds are then evaluated for biological activity through docking simulations, QSAR modeling, or wet-lab testing. Diversity is crucial in this process: selecting redundant compounds from dense chemical regions increases cost and reduces the chance of identifying novel actives. Our framework provides a fast and scalable solution for quantifying and enforcing diversity constraints at this scale.

### B. Diversity Score as a Penalty Term

Let $\mathcal{L} \subseteq \mathcal{C}$ denote a selected subset of molecules from a large chemical library $\mathcal{C}$, and let $D_i$ be the diversity score assigned to molecule $i$, computed as

$$D_i = \frac{1}{n_{\text{cell}(i)}}, \tag{6}$$

where $n_{\text{cell}(i)}$ is the number of molecules residing in the same hypercube as molecule $i$. It is equivalent to (4) but emphasizes $i$ instead of $k$. Lower values of $n$ (i.e., sparse regions) yield higher diversity scores. To integrate this into a compound selection problem, we define an objective function that balances application-specific utility (e.g., predicted activity) with a penalty for redundancy:

$$\max_{\mathcal{L} \subseteq \mathcal{C}} \sum_{i \in \mathcal{L}} \left( \text{Score}_i - \lambda \cdot \text{Penalty}_i \right), \tag{7}$$

where:

- Score$_i$ may be a predicted docking affinity, QSAR score, or even a binary screening hit,
- Penalty$_i = \frac{1}{D_i} = n_{\text{cell}(i)}$ is proportional to local density,
- $\lambda \geq 0$ is a tunable regularization parameter controlling the trade-off between activity and diversity.

This formulation ensures that molecules in dense regions are penalized unless they exhibit exceptionally high predicted activity, while molecules in sparse regions are promoted due to their uniqueness.

### C. Optimization Strategy

Due to the scale of modern chemical libraries, exact subset optimization is computationally infeasible. However, the additive structure of the objective function enables greedy or streaming approximations. A simple heuristic selection process proceeds as follows:

1) Compute $D_i$ for all molecules using our MapReduce framework.
2) For each molecule, compute the adjusted utility:

$$U_i = \text{Score}_i - \lambda \cdot n_{\text{cell}(i)} \tag{8}$$

3) Sort molecules by $U_i$ and select the top-$k$ as the final subset $\mathcal{L}$.

This scoring-based selection is fully compatible with distributed systems and can be efficiently implemented using Spark's sort and filter operations.

### D. Case Study Example

To illustrate the effectiveness of this approach, consider a virtual screening task for kinase inhibitors. Starting with a vendor library of 100 million compounds and using a pretrained QSAR model to predict activity scores Score$_i$, we apply our diversity-aware selection pipeline with $\lambda = 0.5$. The resulting subset of 10,000 compounds includes structurally novel scaffolds that span the chemical space more evenly than standard top-score-only selections. A visual inspection of the descriptor space in Figure 6 confirms that the chosen compounds avoid overrepresentation in dense clusters and instead include candidates from sparsely populated, potentially novel regions.

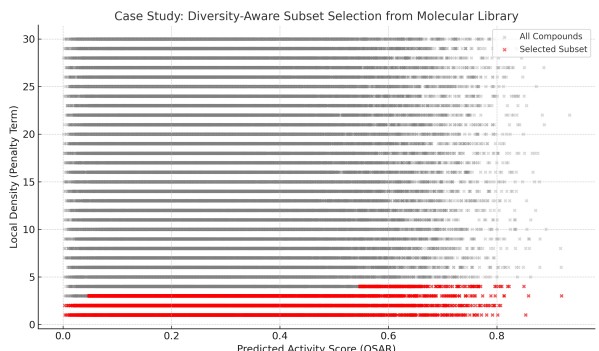

Fig. 6: Case Study: Diversity-Aware Subset Selection from Molecular Library.

## VIII. Conclusion

We present a scalable, cloud-ready framework for diversity analysis of ultra-large molecular datasets, motivated by the demands of modern cheminformatics and computational drug discovery. By partitioning high-dimensional descriptor spaces into hypercubes and leveraging a three-stage MapReduce pipeline on Apache Spark, our method efficiently approximates structural novelty through local density estimation—bypassing the prohibitive cost of pairwise distance calculations. The approach enables diversity-aware compound prioritization across chemical libraries exceeding hundreds of millions of candidates, with strong performance demonstrated in runtime, memory usage, and resolution of sparse regions.

Beyond cheminformatics, this framework offers broad applicability to other biomedical domains that rely on high-dimensional data representations, such as genomic feature selection, phenotypic clustering, and biomarker discovery. The algorithm's linear scalability, modular design, and compatibility with public cloud platforms make it well-suited for integration into cheminformatics screening pipelines where diversity, interpretability, and computational efficiency are critical. Our results illustrate how big data principles—combined with biomedical insight—can transform the tractability of diversity quantification in real-world applications.

Future work will explore extensions to adaptive binning strategies, incorporation of domain-specific ontologies, and integration with multi-objective optimization in personalized medicine and drug repurposing workflows.

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
