# OpenReview forum: "Scalable Diversity-Aware Feature Scoring for Biomedical Big Data via Hypercube-Based Density Estimation"
_IEEE.org/EMBS/BHI/2025/Conference — BHI 2025_

### Official Review · Reviewer_L2Wa · 2025-06-27
**Scalable Diversity-Aware Feature Scoring for Biomedical Big Data via Hypercube-Based Density Estimation**

**Confidence:** 4
**Clarity Of Writing:** excellent
**Clinical Significance:** great
**Methodological Novelty:** excellent
**Overall Rating:** 7

**Experiments And Results:**

good

**Questions For The Authors:**

I have no further questions for the authors.

**Strengths:**

The paper tackles the important issue of efficiently quantifying the molecular diversity from massive databases that traditional methods struggle with. Improvement in this area holds promise to significantly speed up drug discovery and, as such, this paper has a strong translational impact. The paper is well written and clear, even for a reader who does not have a strong background in the field. The authors provide a robust description and reasoning for their approach and the comparison with standard methods highlights the presented method's novelty and promise. The direct description of how this framework could be applied to molecular screening is a strong component of the work.

**Summary Of The Paper:**

The paper presents a novel, computationally efficient method to characterize molecule diversity. The method partitions the descriptor space into a hypercube grid and computes each point's diversity score relative to the number of points within its grid. This setup allows for the processing of each hypercube grid to be performed independently in a parallel fashion, drastically improving computational efficiency relative to standard methods, such as pairwise distance calculations. The authors compare their framework with existing tools and present a potential application in molecular screening and diversity-based selection of molecules.

**Weaknesses:**

The paper mentions a comparison to standard methods in section VI.E. A figure visualizing the results from this comparison would more clearly show the strengths of the presented framework.

While the experimental validation is strong, a validation with real molecular data as opposed to synthetic data would further strengthen the paper. Adding this would strengthen the work but is not necessary if time or complexity does not allow.

---

### Official Review · Reviewer_7rv9 · 2025-07-16
**back arrowBack to Tasks Scalable Diversity-Aware Feature Scoring for Biomedical Big Data via Hypercube-Based Density Estimation**

**Confidence:** 3
**Clarity Of Writing:** good
**Clinical Significance:** good
**Methodological Novelty:** good
**Overall Rating:** 7

**Experiments And Results:**

good

**Questions For The Authors:**

Could hybrid approaches (e.g., applying pairwise refinement within hypercubes) improve diversity accuracy without compromising scalability?

**Strengths:**

Scalability and Efficiency
Practical Implementation
Generality and Applicability
Interpretability and Simplicity

**Summary Of The Paper:**

The paper presents a scalable framework for diversity-aware feature scoring in ultra-large biomedical datasets, focusing on cheminformatics and virtual screening applications. Traditional diversity metrics relying on pairwise distance computations are computationally prohibitive at this scale. To overcome this, the authors propose a hypercube-based density estimation approach implemented via a three-step MapReduce algorithm on Apache Spark. Each molecule is embedded in a normalized high-dimensional descriptor space, assigned to a hypercube, and scored inversely to local cell occupancy, providing a proxy for structural novelty without pairwise calculations. Experiments on synthetic datasets of up to 200 million molecules demonstrate linear runtime scaling, stable memory usage, and applicability for diversity-constrained subset selection, with potential extensions to other biomedical domains such as genomics and phenotypic clustering.

**Weaknesses:**

The choice of bin size (hypercube granularity) significantly affects performance and accuracy, but the paper does not provide an adaptive strategy or theoretical guidance for optimal selection.
Comparison with classical diversity selection methods (e.g., RDKit MaxMin) is minimal and only on very small subsets; no benchmarking against other scalable or approximate diversity algorithms.

---

### Official Review · Reviewer_Xx43 · 2025-07-18
**Scalable Diversity-Aware Feature Scoring for Biomedical Big Data**

**Confidence:** 4
**Clarity Of Writing:** great
**Clinical Significance:** good
**Methodological Novelty:** great
**Overall Rating:** 7

**Experiments And Results:**

good

**Questions For The Authors:**

1. Your scalability results on synthetic data are excellent. Have you considered validating the framework's performance on a large, publicly available, real-world chemical library (e.g., a multi-million compound subset from ZINC or ChEMBL) to demonstrate its effectiveness in handling the specific statistical properties of actual molecular descriptors?
2. The number of bins ($m$) is a critical hyperparameter. Could you provide a more detailed sensitivity analysis? For example, how does the rank-ordering of compounds by diversity score change as $m$ is varied, and how does this impact the composition of a final selected subset in your case study?
3. In your qualitative comparison with RDKit's MaxMin algorithm, you note that your method "slightly favors edge regions". Could you elaborate on this observation? Do you view this as an advantage (i.e., better exploration of under-sampled chemical space) or a potential bias of the hypercube-based method?

**Strengths:**

-  The paper tackles a well-defined and significant bottleneck in modern drug discovery: the inability of traditional diversity quantification methods to handle ultra-large chemical libraries. The proposed solution is specifically designed to address this challenge.
- The hypercube partitioning approach is an intelligent and efficient way to approximate local density without incurring the prohibitive $O(N^2)$ cost of distance matrix calculations. The implementation as a three-step MapReduce algorithm in Spark is a natural fit for distributed computing, and the demonstrated linear runtime is a significant achievement.
- The framework is built using industry-standard big data technologies (Apache Spark, MapReduce) and is explicitly designed for cloud deployment on platforms like AWS. This makes the method practical and immediately applicable for researchers and companies working with large-scale biomedical data.
- The paper excels in demonstrating the practical utility of its diversity score. The section on diversity-constrained selection clearly outlines how the score can be used as a penalty term to balance bioactivity and structural novelty, a common goal in virtual screening. The case study example effectively illustrates this application.
- While the primary motivation is cheminformatics, the authors correctly highlight that the density-based scoring framework is domain-agnostic. It can be readily applied to other high-dimensional data problems in biomedicine.

**Summary Of The Paper:**

This paper presents a highly scalable framework for quantifying molecular diversity in ultra-large datasets, a critical task in cheminformatics and early-stage drug discovery. The authors address the severe computational limitations of classical diversity metrics, such as pairwise distance calculations, which are infeasible for modern chemical libraries containing hundreds of millions of compounds. The proposed solution is a fast, density-based scoring method that eliminates the need for pairwise comparisons. The core idea is to partition a high-dimensional molecular descriptor space into a grid of discrete hypercubes. Each molecule's diversity score is then calculated as the inverse function of the local cell occupancy, which is the number of other molecules that fall into the same hypercube. This provides a computationally efficient proxy for structural novelty. The entire framework is designed as a three-step MapReduce algorithm implemented in Apache Spark, making it inherently parallel and cloud-ready. The authors validate the method's performance on synthetic datasets of up to 200 million molecules, demonstrating near-linear runtime scaling and stable memory usage on a 10-node cloud cluster. The paper also provides a clear use case, showing how this diversity score can be integrated as a penalty term in a diversity-constrained optimization problem for virtual screening. While motivated by cheminformatics, the authors note that the domain-agnostic framework is generalizable to other biomedical applications, such as genomic feature selection.

**Weaknesses:**

- The core scalability experiments, while impressive, were conducted using synthetic data where descriptor vectors were sampled from a normal distribution. This effectively validates the computational performance (in terms of runtime and memory) but does not demonstrate the method's effectiveness on real chemical descriptor data, which often exhibits complex, non-uniform distributions and strong correlations between features. The comparison to RDKit's MaxMin algorithm was only performed qualitatively on a very small subset of 10,000 compounds.
- The method's outcome is highly dependent on the granularity of the grid, which is controlled by the number of bins per dimension ($m$). The paper mentions this parameter but provides only a brief heuristic for tuning it. A more rigorous sensitivity analysis showing how the diversity scores and the resulting selected subsets change with different values of $m$ would significantly strengthen the work.
-  By design, all molecules within the same hypercube are assigned the same diversity score, which is a coarse approximation that loses fine-grained, intra-cell distance information. While this trade-off enables scalability, a more detailed discussion of its implications on the quality of diversity selection would be beneficial.